# Shame, Guilt, and Self-Consciousness in Anorexia Nervosa

**DOI:** 10.3390/jcm11226683

**Published:** 2022-11-11

**Authors:** Matteo Panero, Paola Longo, Carlotta De Bacco, Giovanni Abbate-Daga, Matteo Martini

**Affiliations:** Eating Disorders Center for Treatment and Research, Department of Neuroscience, University of Turin, Via Cherasco 11, 10126 Turin, Italy

**Keywords:** anorexia nervosa, guilt, shame, self-conscious emotions, psychotherapy

## Abstract

Background: the role of self-conscious emotions (SCE) such as shame and guilt in eating disorders (ED) has been systematically studied only in recent years, but it is still debated. This study aims to investigate the role of SCE in anorexia nervosa (AN), evaluating the role of self-consciousness. Methods: fifty-five individuals with AN and seventy-four healthy controls (HC) were enrolled and completed a battery of tests evaluating the proneness to feel shame and guilt, as well as comparing self-consciousness, eating, and general psychopathology. Results: individuals with AN showed a higher proneness to shame. Shame was correlated with body dissatisfaction and drive for thinness, which are core symptoms in AN, after controlling for scores of depression and anxiety. Proneness to guilt seemed to be less correlated with eating and body symptomatology, but it appeared to have a negative correlation with binge-purging symptoms. Furthermore, proneness to shame was independent of guilt or self-consciousness and the two groups did not differ regarding public and private self-consciousness. Conclusions: shame is an important and independent factor in AN. Future research may offer progress in the development of shame-focused therapies.

## 1. Introduction

Anorexia nervosa (AN) is a mental disorder that typically develops during puberty and adolescence and entails serious consequences [1]. Despite the advances made in multimodal interventions for AN in the last decades, outcomes are still variable and relapse rates are reported as high [2,3]. According to theoretical conceptualizations and research evidence, emotional states and their regulation play an important role in the onset and maintenance of AN [4,5]. During the last ten years, the research on emotional dysregulation in psychiatric disorders has been growing [6,7], showing that both alexithymia (i.e., the inability to identify and describe emotions) and the use of maladaptive strategies in emotional regulation correlates with psychopathology (for reviews see [8,9,10]). Recent studies on the topic highlighted that individuals with AN rely on maladaptive schema and emotion regulation strategies [11] and show poorer emotional awareness when compared with healthy controls (HC) [12,13,14,15,16,17,18].

Classically, emotions have been categorized into the so-called primary emotions (i.e., anger, fear, sadness, disgust, happiness, and surprise) and the supposedly more elaborate secondary emotions [19]. These latter emotions, and in particular self-conscious emotions (SCE), require a sense of self and cognitive skills that depend on the person’s own experience, expectations, and beliefs. They relate to standards, rules, goals, or expectations of a society or a group [19,20]. Among these, shame and guilt are two transdiagnostic emotional traits that have been studied in different forms of psychopathology [21,22,23,24]. In Tangney’s theoretical framework [19,25] guilt is conceptualized as the consequence of a negatively evaluated behavior and implies internal, specific, controllable, attributions. Whereas in shame the entire self is scrutinized and negatively evaluated and a sense of being small and worthless is conveyed. In other words, guilt entails a sense of tension, remorse, and regret but does not affect one’s core identity, unlike shame. The two SCE involve distinct sets of actions: shame is associated with a desire to disappear, to hide, or to escape, while feeling a sense of guilt is associated with a desire to repair or undo harm [26,27]. 

SCE are thought to depend on the so-called self-reflective functions. Self-consciousness is the tendency to focus attention on one’s own thoughts, feelings, and attitudes and the proneness to focus on publicly observable aspects of self-qualities. Elevated self-focused attention has been demonstrated in most forms of psychopathology [28] and there is an increasing number of studies investigating the association between dimensions of the self-concept (including global self-esteem, body size estimation disturbances, emotional self-regulation, and attitudes toward body image) and eating disorder symptomatology [29,30,31]. Current research however has rarely explored self-consciousness and SCE together in individuals with AN. For instance, it would be relevant to assess if individuals with AN are more prone to experiencing SCE due to heightened self-consciousness in comparison to healthy individuals.

Previous research has suggested that the role of SCE in personal and interpersonal identity may explain key characteristics of individuals with AN such as self-criticism, fear of being seen, ascetic behaviors, and perfectionism [32]. Bodily shame, especially for those parts considered “fat” or disproportionate, is predictive of eating disturbance in AN [33] and gaining control over body shape could be a way to reduce shameful feelings [13]. Furthermore, shame proneness in individuals with AN relates to a drive for thinness [34], emotional avoidance, ambivalence, and feelings of unworthiness [35], which have been found to relate to interoceptive deficits and low self-esteem [36]. Other studies suggested that dietary restriction and excessive physical activity in AN could have the function of reducing guilt [37] and that both shame and guilt could elicit binge-purging symptoms [38,39]. Moreover, the impairments in social interactions experienced by individuals with AN [40,41] could be related to the crucial role of SCE in interpersonal contact [26]. Furthermore, there is a lack of studies in the ED field that addresses shame and guilt together, and no study has yet controlled for guilt when measuring shame or vice versa [4].

Lastly, shame and guilt have been shown to co-occur with the development of anxiety and—especially in adolescents—depressive symptoms [24,42]. Despite the relevance of anxiety and depressive symptomatology in AN, the connections between shame and guilt and anxiety and depressive symptoms in this disorder are still understudied [4,5]. It would be clinically relevant to ascertain if the presence of an anxiety and depressive comorbidity had a direct relationship on increased levels of guilt and shame in the course of AN.

Therefore, grounded on these premises, with this study we aimed:

(a)to evaluate self-conscious emotions (guilt and shame) and self-consciousness in individuals with anorexia nervosa by comparing them with a group of healthy individuals;(b)to evaluate if depressive and anxiety symptoms or other confounding factors can explain differences in experiences of shame and guilt among the two samples (including controlling for guilt scores when evaluating shame scores and vice versa);(c)to explore self-consciousness and self-conscious emotions in different AN subtypes (restricting or ANR and binge-eating/purging subtype or ANBP);(d)to ascertain the existence of a correlation between self-consciousness and self-conscious emotions and psychometric and clinical variables and identify the emotional variables that independently correlate with the diagnosis of AN.

We expected to find proneness to shame as prominent in AN individuals in comparison to HCs, whereas we had no specific predictions regarding the role of proneness to guilt. We also expected to find a direct relationship between self-consciousness and both clinical severity and shame and guilt levels. Furthermore, we expected shame and guilt to be dependent on anxiety and depressive symptoms and to find proneness to shame and guilt as high in both subtypes of AN.

## 2. Materials and Methods

The study sample consisted of fifty-five AN outpatients and inpatients (hospitalization and day treatment) who consecutively sought treatment at the Eating Disorders Center of the University of Turin, Italy. Individuals treated in this specialized ED unit are referred from the whole Piedmont region and have often failed previous treatments or have been experiencing severe medical conditions.

The inclusion criteria for this study were: (1) full-criteria diagnosis of anorexia nervosa and (2) age between 18 and 50 years old. Exclusion criteria were: (1) current diagnosis of psychotic, bipolar, or substance use disorders or (2) intellectual deficits. The diagnosis was assessed by an experienced psychiatrist using the Structured Clinical Interview for DSM-5 Axis-I Disorders (SCID-I). Forty-three individuals were suffering from anorexia nervosa restricting type (AN-R) and twelve from anorexia nervosa binge-eating/purging type (AN-BP). Fifty-four were female and one was male. Eighteen individuals (32%) were suffering from other psychiatric illnesses (16 were diagnosed with major depression disorder and 2 with anxiety disorder); twenty-three (41%) individuals were under psychotropic medications (antidepressants and anxiolytics). 

Seventy-four gender-matched HCs were recruited from visiting students and collaborators via the snowball sampling method; they were then interviewed in person to measure their BMI and to ascertain the following inclusion criteria: (1) no lifetime history of mental disorders according to DSM-5 criteria [43]; (2) no use of medications; (3) no current or lifetime organic illness as assessed per clinical interview; (4) age > 18 and <55 years old. 

All participants were asked to complete the following self-report questionnaires: Self-Consciousness Scale (SCSR), Test of Self-Conscious emotions (TOSCA), Toronto alexithymia Scale (TAS), Beck Depression Inventory I (BDI), State and Trait Anxiety Inventory (STAI), and Eating Disorder Inventory–2 (EDI-2). No one refused to complete the questionnaires. All participants provided written informed consent for this study. This study was approved by the Ethical Committee of the Department of Neuroscience of the University of Turin, Italy. All procedures were conducted in accordance with the latest version of the Declaration of Helsinki. 

### 2.1. Measures

Test of Self-Conscious Affect 3: The Italian version of the Test of Self-Conscious Affect (TOSCA) [44,45] is a scenario-based measure that assesses individual differences to the degree to which people are prone to experience shame and guilt across a range of hypothetical situations involving failures or transgression. Cronbach’s alphas for adult and student samples have been estimated as 0.76 and 0.76 for shame and 0.66 and 0.60 for guilt [46]. Reported test-retest reliabilities for the TOSCA tend to be 0.85 for shame and 0.74 for guilt [33,34].

The Self-Consciousness Scale-Revised (SCS-R): The SCS-R [47] is a revised version of the SCS [48]. It consists of 23 items measured on a five-point Likert scale, which were divided into three dimensions: private self-consciousness (9 items), public self-consciousness (7 items), and social anxiety (6 items). A reliability score was sufficient (α = 0.73 and 0.89 for test-retest) [49]. Cronbach’s alpha was found to be 0.75 for private self-consciousness and 0.84 for public self-consciousness [26]. 

Eating disorder inventory-2 (EDI-2) [50]: The EDI-2 is a self-report inventory that measures eating psychopathology with eleven subscales evaluating symptoms and psychological correlates of eating disorders. All EDI-2 subscales showed significant test-retest correlations ranging from 0.81 to 0.89 and Cronbach’s alpha from 0.82 to 0.93 [51].

Beck Depression Inventory I (BDI) [52]: The validated Italian version [53] of a 13-item self-report questionnaire used to measure self-reported depressive symptoms. BDI internal consistency has a mean coefficient alpha of 0.86 for psychiatric patients and 0.81 for non-psychiatric individuals. Test-retest reliability ranged from 0.73 to 0.96 [54]. 

State-Trait Anxiety Inventory (STAI) [55]: The validated Italian version [56] of a 20-item instrument to self-report state anxiety, a temporary condition experienced in specific situations, and trait anxiety, a general tendency to perceive situations as threatening. Internal consistency (Cronbach’s alpha) varies from 0.86 to 0.95; test-retest reliability coefficients vary from 0.65 to 0.75 over a 2-month interval [55]. 

Body Shape Questionnaire (BSQ) [57]: A 34-item self-report questionnaire evaluating body image and body dissatisfaction in the last two weeks. The test re-test reliability has been evaluated as good (r = 0.88) for all 34 items [58]. 

The Toronto Alexithymia Scale -20 (TAS) [59]: A 20-item self-report questionnaire measuring alexithymia; it assesses three scales: difficulty identifying feelings, difficulty describing feelings, and externally oriented thinking. A fourth scale is the sum of the former ones and measures the global alexithymia score. The test demonstrates good internal consistency (Cronbach’s alpha = 0.81) and test-retest reliability (0.77, *p* < 0.01) [60]. 

### 2.2. Statistical Analysis 

Data collected from individuals with AN and HC were analyzed using SPSS, the Statistical Package for Social Science (Version 24 for Mac). First, data were inspected and descriptive statistics were calculated for all participants.

To address Aim (a), Student *t*-tests were conducted comparing AN and HC groups. 

To address Aim (b), several one-way ANCOVAs were conducted using groups (AN and HC) as fixed factors, proneness to shame as the dependent variable and BMI, age, trait anxiety (investigated through STAI), depression (investigated using BDI), alexithymia (investigated by TAS), and guilt (investigated by TOSCA-3) as covariates. 

To address Aim (c), one-way ANOVA was conducted comparing ANR, ANBP, and HC, and subsequently was conducted as a post-hoc test (Tukey’s HSD-honestly significant difference).

To address Aim (d), correlations between the TOSCA-3 and SCSS-R scales and EDI-2, BSQ, BDI, TAS, scales, and psychometric information (BMI, duration of illnesses) were assessed with Pearson’s r coefficient. All correlations were two-tailed. Finally, to evaluate the emotional variables that independently correlate with belonging to the AN group, a logistic regression analysis using the stepwise Wald method was conducted: the dependent variable was group (AN or HC) and the independent variables were proneness to shame, proneness to guilt, public self-consciousness, private self-consciousness, alexithymia (TAS). 

A *p* level of 0.05 was considered significant.

## 3. Results

### 3.1. Clinical and Sociodemographic Characteristics 

Individuals with AN and HCs did not differ in age and were significantly different in BMI as shown in Table 1. Furthermore, the AN sample was composed of individuals who were severely underweight and had experienced a long-duration of illness. Individuals under pharmacological treatments did not differ from those who did not use medications regarding baseline characteristics and psychometric data from TOSCA-3 and SCS-R (data not shown). Mean EDI-2 scores in the HC group were below the cut-offs for EDs [61].

### 3.2. Comparison between AN and HC (Aim a)

In Table 2, the comparison between individuals with AN and HC concerning TOSCA-3 and SCS-R is presented. AN subjects showed higher shame proneness and did not differ from HCs in self-consciousness.

### 3.3. Difference in Proneness to Shame in AN and HC Groups: The Role of Depression, Anxiety, and Confounding Factors (Aim b)

In Table 3, proneness to shame is individually examined to assess the influence of confounding factors. As shown, proneness to shame is independent of anxiety and depressive symptoms, alexithymia, guilt, and self-consciousness, but is influenced by BMI. 

### 3.4. Proneness to Shame and Proneness to Guilt in AN Subtypes (Aim c)

The role of shame and guilt was then studied separately in AN-R and AN-BP samples. Table 4 shows a higher level of shame in the AN-BP sample and a higher level of guilt in the AN-R sample, but no significance was achieved.

### 3.5. Correlations and Regression Analysis in the AN Sample (Aim d)

In Table 5, the results are shown of the correlations that have been performed between TOSCA-3 and SCSR and psychopathological and clinical variables in the AN population. For sake of concision, only significant data are shown.

Lastly, a regression analysis was conducted to evaluate which variable among TOSCA-3, TAS, and STAI, better describes the probability to be either AN or HC. The best-fitting model included TAS Total (R2 = 0.274) and the second-best-fitting model included TAS Total and proneness to shame (R2 = 0.307). All other variables were excluded from the model. The model is described in Table 6.

## 4. Discussion

In this study, we evaluated self-conscious emotions in individuals with AN in comparison to HCs and explored their significant psychopathological correlates. Main findings emerged: (1) proneness to shame was higher in individuals with AN than in HCs, and the significant difference was maintained when evaluating confounding factors; (2) proneness to guilt was not different among AN and HC; (3) individuals with AN and HCs did not significantly differ in public and private self-consciousness, thus suggesting that proneness to shame characterizes AN better than high self-consciousness; (4) proneness to guilt did not affect proneness to shame, which is in line with the independent role of these two emotions in AN; (5) proneness to shame was associated with the core clinical variables of drive for thinness, body dissatisfaction, and discomfort with bodily concerns and remained a significant correlate to AN diagnosis in the regression analysis; and (6) there were no significant differences in shame and guilt among AN subtypes. 

### 4.1. The Role of Shame

Individuals with AN showed higher scores in proneness to shame in comparison to HCs even when assessing depressive and anxiety symptomatology as covariates, and this difference was confirmed when the confounding role of age and duration of illness were considered. These associations have rarely been evaluated in AN [34,62], and our findings suggest that feelings of shame can be considered relevant for AN psychopathology independently of depressive and anxiety comorbidity. The significant correlations between proneness to shame and the key dimensions of body dissatisfaction, body shape concerns, and—more weakly—drive for thinness suggest that the tendency to experience pervasive feelings of worthlessness elicited by shame proneness could be linked with greater difficulties in evaluating body image and shape in individuals with AN. In understanding the processes related to starvation and food denial typical of AN, proneness to shame could explain this core area of eating psychopathology. 

Furthermore, proneness to shame was not independent of BMI; therefore, further studies with larger sample sizes should evaluate the correlation between malnutrition and shame. 

Finally, proneness to shame does not correlate with the duration of illness; thus, it does not seem to change or attenuate after a long course of illness and is not just a consequence of long duration of illness or demoralization or depressive symptoms, which often emerge in AN [63]. Therefore, proneness to shame may represent a stable element to target during treatment as well as an emotional focus in psychotherapy. In fact, despite the role of emotions in ED pathogenesis highlighted by theoretical models [64], still there are few specific treatments with emotional focuses in AN. For instance, in enhanced cognitive behavioral therapy, only mood intolerance is considered a focus. Addressing shame in psychotherapy could be of value for two main reasons: (1) because shame is a social emotion, working on reducing shame may target relational impairment in AN; (2) body shame is a risk and maintenance factor for AN: individuals with AN themselves reported overcoming body shame as a treatment target [65].

### 4.2. Shame and Self-Consciousness in AN

Regarding the potential role of increased self-consciousness in AN psychopathology, the lack of differences in public and private self-consciousness as evaluated by the SCRS questionnaire suggest that AN individuals do not seem to scrutinize themselves more than HCs. This finding leads us to conclude that individuals with AN could be better characterized by shame than by heightened concerns regarding their self-images in their internal and social worlds. This conceptualization seems to be in line with theories that claim that the core problem in AN is the inner experience of incompetence and indignity rather than the judgment based on body shape observation [41,66]. Indeed, differences in proneness to shame between individuals with AN and HCs remained significant even when we considered the potential influence of public and private self-consciousness. This is noteworthy, as it is expected that a higher proneness to shame entails a major tendency to control the self or vice versa. This result could be the consequence of two opposite behaviors in AN: significantly high frequency of body checking [67] but also avoidance of somatic negative stimuli [68]. Furthermore, in the regression analysis, only proneness to shame and alexithymia were correlated with AN, whereas the scales of SCSR were excluded from the model. In contrast with Zucker [69], who claimed that self-focused attention is a core aspect of AN, our data suggest that the differences between AN and HC concerning self-conscious emotions do not derive from differences in self-perception and self-conscious evaluation. Negative emotions, especially shame, are more important in defining AN psychopathology than self-focused attention. 

### 4.3. Guilt-Free Shame

Another interesting finding is the independence between proneness to shame and proneness to guilt. As Blythin et al. [4] stated “the potential impact of confounding variables [has been] often overlooked […] No papers controlled for guilt when measuring shame, or shame when measuring guilt”; furthermore, Blythin and colleagues sustain that “there is a lack of studies that address [shame and guilt] together”. In our study, the proneness to shame remained significantly different between individuals with AN and HCs when clinical data (TAS, STAI, BDI) were accounted for as covariates and when using guilt as a covariate. This new datum may be important in solid differentiation of the role of shame and guilt in AN [70]. “Guilt-free shame”, using a concept of Tangney [71], is associated with poor mental health and diminished interpersonal capacity in healthy individuals; thus, it could be of interest when addressed as a focus in treatments.

### 4.4. The Role of Guilt

It is worth noting that individuals in either the AN and HC groups showed no differences in proneness to guilt. This finding is in fact in agreement with previous literature, which has given major importance to the role of shame and its psychopathological correlates in comparison to guilt [72,73]. We hypothesize that guilt could be better studied in close temporal connection with specific symptomatologic characteristics, similarly to what Stevenson et al. [74] have done for the study of emotions before and after loss of control in eating in a non-clinical sample. 

In our clinical sample, the proneness to guilt was significantly and negatively correlated with bulimic symptoms, suggesting that higher levels of binge-purging symptoms are related to less intense feelings of guilt. This finding appears to disagree with the literature that identified guilt as prodromal to binge eating [75] and increased after a long course of bulimic symptoms [76]. However, this could be explained by the characteristics of the sample, which consisted exclusively of individuals with AN and did not include those with bulimia nervosa or binge eating disorder. A negative correlation may be indicative of a regulatory function of guilt for binge-purging behaviors; in individuals with AN, the proneness to feel guilt (as opposed to shame) could be a sign of enhanced behavioral control. This could be confirmed by the larger component of ANR individuals in the sample and could be interesting in shedding some light on the difference between binge-purging symptoms in AN and BN individuals. 

### 4.5. Shame and Guilt among AN Subtypes

Furthermore, when comparing individuals with ANBP subtype and ANR subtype no significative differences emerge, but ANBP individuals showed higher shame proneness scores. Bauer et al. [77] also found that shame was more present in those suffering from bulimic symptoms, and Duarte & Pinto-Gouveia [78] found that shame significantly impacts binge eating, even when controlling for depressive symptoms, in subjects with binge eating disorders. These data are a stimulus for further studies that can differentiate between binge-purging subtype and BN-identifying trigger factors of binge-purging symptoms and differentiate between the emotional aspects prevailing in different diagnoses.

### 4.6. Limitations

Despite interesting findings, some limitations should be acknowledged as well. The sample is modest in size and the individuals recruited were at different stages of their clinical course (i.e., a duration of illness ranging from 1 to 19 years) and presented different degrees of severity. Except for depressive and anxiety symptoms, psychiatric comorbidity was not considered. Furthermore, correction for multiple comparisons has not been applied. Finally, the study of emotional variables is still difficult, and integrating self-reported questionnaires with other instruments such as clinician-administered interviews and ecological assessments could provide more detailed insights into emotional experiences and related behaviors.

## 5. Conclusions

In conclusion, this study highlights the importance of proneness to shame in anorexia nervosa and shows that proneness to guilt has a less clear role and could relate to diminished binge-purging symptoms in AN. Furthermore, the proneness to these two SCE is not merely a consequence of the extreme self-control and self-scrutinizing typical of AN. Shame emerges as a feature central to AN even when controlling for the frequently co-occurring psychopathological symptoms of depression and anxiety. Finally, the presence of shame in the sample of individuals with AN was not influenced by the proneness to guilt. 

As stated in clinical guidelines, psychotherapy is a crucial treatment for AN [79]. In some manualized treatment models [80], specific emotional aspects are addressed with ad hoc treatment stages. In line with the data presented we suggest that specific stages of psychological treatments in AN should address shame and its consequences in eating psychopathology and social behavior.

## Figures and Tables

**Table 1 jcm-11-06683-t001:** Clinical characteristics.

		AN (N = 55)	HC (N = 73)	*T*-test	
BMI	Mean	15.69	20.28	T	14.79
	SD	1.77	1.65	p	0.001 *
Age	Mean	22.54	23.42	T	1.38
	SD	4.92	1.95	p	0.171
Age of Onset	Mean	16.78			
	SD	2.83			
Years of Illness	Mean	6.00			
	SD	4.41			

BMI: body mass index; AN: anorexia Nervosa; HC: healthy controls. ***** = *p* < 0.05.

**Table 2 jcm-11-06683-t002:** AN and HC comparing Shame Guilt and Self-Consciousness.

			AN (N = 55)	HC (N = 73)	*T*-test	
SCSR	Public self-consciousness	Mean	17.85	16.30	T	−1.89
		SD	4.29	4.55	p	0.062
	Private self-consciousness	Mean	14.19	13.22	T	−1.49
		SD	3.52	3.48	p	0.138
TOSCA-3	Proneness to shame	Mean	53.66	47.27	T	−3.10
		SD	12.70	9.58	p	0.002 *
	Proneness to guilt	Mean	63.94	61.90	T	−1.40
		SD	8.38	7.51	p	0.165

BMI: body mass index; AN: anorexia nervosa, HC: healthy controls; TOSCA-3: Test of Self-Conscious Affect; SCSR: Self-Consciousness Scale Revisited. ***** = *p* < 0.05.

**Table 3 jcm-11-06683-t003:** Analysis of covariance evaluating Proneness to shame with covariate.

		f	*p*
Covariate	Trait anxiety (STAI)	8.22	0.005 *
Dependent	Proneness to shame	4.21	0.043 *
Covariate	Depression (BDI)	2.90	0.92
Dependent	Proneness to shame	5.80	0.018 *
Covariate	Alexithymia (TAS)	7.47	0.007 *
Dependent	Proneness to shame	4.51	0.036 *
Covariate	BMI	1.28	0.260
Dependent	Proneness to shame	0.817	0.368
Covariate	Age	0.041	0.840
Dependent	Proneness to shame	1.805	0.015 *
Covariate	Proneness to guilt (TOSCA-3)	21.77	0.001 *
Dependent	Proneness to shame	7.50	0.007 *
Covariate	Public Self-Consciousness (SCSR)	4.47	0.037 *
Dependent	Proneness to shame	7.48	0.007 *
Covariate	Private Self-Consciousness (SCSR)	3.00	0.086
Dependent	Proneness to shame	8.18	0.005 *

BMI: body mass index; TOSCA-3: Test of Self-Conscious Affect; SCSR: Self-Consciousness Scale revisited; BDI: Beck Depression Inventory; STAI: State-Trait Anxiety Inventory; TAS: Toronto Alexithymia Scale. Degree of freedom: 127. ***** = *p* < 0.05.

**Table 4 jcm-11-06683-t004:** Shame and guilt in AN Subtypes. Post hoc test.

		AN-R (N = 43)	AN-BP (N = 12)	HC (N = 73)	Tukey’s HSD
Proneness to shame (TOSCA-3)	Mean	52.85	56.41	47.27	HC < AN-R & AN-BP *
	SD	13.40	9.97	9.58	
Proneness to guilt (TOSCA-3)	Mean	64.95	60.50	61.90	No significant differences
	SD	7.28	11.05	7.51	

AN-R: anorexia nervosa restricting subtype; AN-BP: anorexia nervosa binge-eating/purging subtype; HC: healthy controls; TOSCA-3: Test of Self-Conscious Affect; HSD: honestly significant difference. Degree of freedom: 54. ***** = *p* < 0.05.

**Table 5 jcm-11-06683-t005:** Pearson correlations in AN individuals.

Variable	*r*	*p*	Variable
Public self-consciousness (SCSR)	0.32	0.019 *	Proneness to guilt (TOSCA-3)
	0.37	0.007 *	BMI
	0.40	0.013 *	Body shape concerns (BSQ)
Private self-consciousness (SCSR)	0.31	0.026 *	Proneness to guilt (TOSCA-3)
Proneness to shame (TOSCA-3)	0.30	0.044 *	Drive for thinness (EDI-2)
	0.35	0.019 *	Body dissatisfaction (EDI-2)
	0.34	0.037 *	Body shape concerns (BSQ)
Proneness to guilt (TOSCA-3)	−0.295	0.049 *	Bulimia (EDI-2)

BMI: body mass index; TOSCA-3: Test of Self-Conscious Affect; SCSR: Self-Consciousness Scale revisited; BDI: Beck Depression Inventory; BSQ: Body Shape Questionnaire; EDI-2: Eating Disorders Questionnaire 2. Degree of freedom: 54. ***** = *p* < 0.05.

**Table 6 jcm-11-06683-t006:** Stepwise Logistic Regression Model and Statistics for Dependent Variable AN or HC.

Model	Unstandardized Coefficients		Standardized Coefficient Beta	t	*p*	R2	F
	B	Standard Error					
1						0274	32.016
Constant	−5.439	1.162	0.004	21.919	0.001 *		
Alexithymia (TAS)	0,104	0.23	1.109	20.675	0.001 *		
2						0.307	36.194
Constant	−7.527	1.664	0.001	20.464	0.001 *		
Alexithymia	0.90	0.027	1.055	15.182	0.001 *		
Proneness to shame (TOSCA-3)	0.54	0.023	1.094	3.960	0.047 *		

TOSCA-3: Test of Self-Conscious Affect; STAI: State Trait Anxiety Inventory; TAS: Toronto Alexithymia Scale. Excluded variables: proneness to guilt, public self-consciousness, private self-consciousness, STAI trait and state. ***** = *p* < 0.05.

## Data Availability

The data that support the findings of this study are available from the corresponding author upon reasonable request. The data are not publicly available due to privacy.

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
