# Peer review of "Shame, Guilt, and Self-Consciousness in Anorexia Nervosa"

_jcm, 2022, doi:10.3390/jcm11226683_

Round 1

Reviewer 1 Report

The authors present a case-control study comparing self-conscious emotions (SCE) in patients with Anorexia nervosa and healthy controls. The topic is of interest to those treating AN patients.

Please consider using English language support to correct for grammar mistakes and phrasing.

Introduction

Lines 73/74: Please consider using the common nomenclature restricting pattern and binge-purge pattern.

Methods

The AN sample includes patients with dangerously low body weight and according to the BMI range healthy controls appear to almost meet the criteria of AN as well (BMI of 18.65). Please describe the sample in more details.

You mention that almost half of the AN patients (41%) receive medication. Please describe the type of medication and please also discuss the potential impact on psychopathology.

Please explain why you used the BDI I instead of BDI II.

Results

As shown, proneness to shame is independent from anxiety and depressive symptoms, alexithymia, guilt, and self-consciousness, but is influenced by BMI (Line 177).

Please elaborate in more detail on the association between BMI and proneness to shame.

Please also state how HCs scored in the EDI-2, regarding the low BMI of some participants.

Discussion

When describing the correlations between EDI-2 items and proneness to shame please state that there is only a weak association (r < 0.4).

Please rephrase lines 223-226: “Interestingly, in the clinical sample, proneness to shame showed correlations with drive for thinness, body dissatisfaction and body shape concerns (see table 5), thus the tendency to experience pervasive feelings of worthlessness elicited by shame proneness may be linked with greater difficulties in evaluating image and body shape in AN subjects.”

One would expect differences in test results according to duration of illness and your sample shows quite a large variability (between 2 and 10 years). Please discuss.

Please rephrase lines 247-251: “Data confronted with the statement of Zucker [77], claiming that self-focused attention is a core aspect of AN, suggesting that the noteworthy differences among AN and HC concerning self-conscious emotions it is not due to the difference and the quality of self-perception and self-conscious evaluation and that negative emotions, especially shame, could be more important in defining AN psychopathology than self-focused attention.”

Line 264: “AN sufferers”. Please consider using another term.

The discussion is lacking information on how the results of this investigation should influence treatment of AN, if at all, and how this should be addressed.

Conclusion

Lines 302/309: shame stands on her own/ address shame and his consequence. Please consider using “its”.

Line 307: It is not clear what is meant by “the role of emotion has always been a core of therapies walking-through”

Reviewer 2 Report

This article addresses an interesting and important topic area regarding the role of shame and guilt in anorexia nervosa. While the ideas behind the research are worthwhile and original, I found the article a bit difficult to follow at times and had confusion around statistical decisions used. I also feel that the aims were not clearly followed across all sections. As a result the impact of this research has been a bit obscured. A better logical progression from the aims, to choice of statistical analyses, to reporting of results and then to organisation of the discussion, would help very much. This may require reviewing the aims and revising the statistical methods used. Additionally, significant English language editing would greatly improve the written clarity.

Additional notes:

1. Lines 35 – 37: It is very helpful to have provided a definition of self-conscious emotions, but perhaps these lines could be rephrased or broken down for increased clarity. This is a very long sentence.

2. Please use person-centred language. E.g., instead of “anorexia patients” or “subjects suffering from Anorexia Nervosa” please say “people with anorexia”.

3. Line 47: I am not sure what is meant by the word “borrow” in the following sentence - “shame and guilt borrow self-evaluation, self-criticism… etc”. Perhaps there is a better way to explain what is meant?

4. I think “Restrictor subtype” usually phrased as “restricting subtype”?

5. Line 92-94: Were comorbid psychiatric illnesses assessed with SCID interview as well? It is unusual to have only depression and anxiety as additional diagnoses, and in such a small percentage of participants.

6. Lines 95-98: Were psychometric tests self-report or clinician administered?

7. Statistical analysis section: It was unclear from the description why certain tests were used and for what variables.

8. Statistical analysis section: The stated aims do not clearly line up with the chosen analyses. It would be better to break down the analyses aim-by-aim. E.g., structure as follows: “To address Aim 1, [test type] was used to examine [expected relationship] among [list what variables are being examined]. To address Aim 2…. Etc.”

9. The ANCOVAs or logistic regression were unexpected as it is unclear how these line up with stated aims. Choice of tests needs to be more thoroughly justified and explained.

10. Table formatting: Not sure on whether this is prescribed by the journal format or not, but it seems that commas have been used instead of full stops to signify decimal places? Normally a full stop should be used to avoid confusion.

11. Tables: Use of Σ is unusual? Is this meant to indicate standard deviation (SD)? Sum certainly doesn’t seem right!

12. Results: Again, it would be helpful if the results were presented according to each Aim so that it was clear what result was addressing what aim.

13. Table 4: Should the result for Tukey’s HSD be written as “HC < AN-R, AN-BP”, instead of " HC>AN-R, AN-BP”, as the AN groups had higher scores than HC? Where is the result for Proneness to guilt? If non significant, can at least write “no significant differences” instead of leaving blank.

14. Table 5: A traditional correlation table layout with asterisk to indicate significant results may be more appropriate?

15. There has been no corrections for multiple comparisons applied. Please justify.

16. Discussion: It was sometimes difficult to understand the points being made in the discussion. A more organised structure and heavy editing of English language use would help with this. Again I suggest structuring the discussion to follow and address each of the aims that were proposed in the introduction section.

17. The discussion swaps between specifying AN subtypes or just saying ‘AN’ broadly, so it is harder to determine what results apply to what group. I became a bit confused by this.

18. There are a great many references, perhaps these can be cut down.

Round 2

Reviewer 2 Report

I thank the authors for their clear efforts in responding to the feedback provided. Major changes have been made that address all comments raised, and thus the manuscript is significantly improved. There is now a clear and logical progression from the introduction and aims through all sections of the paper, and I found it much easier to grasp key discussion points which highlight the importance of the findings. This helps reader's interest and understanding of the value of the research.

I have only a few minor suggestions remaining, but otherwise am pleased with the changes made:

1. Aim D could be revised slightly to be more clear and comprehensively explained, as it is still vague and did not seem to clearly lead to the logistic regression.

2. I noticed there were still some minor typographical errors throughout the manuscript, and some areas requiring further English language proofing to improve the clarity of expression. 

Author Response

We thank you for your appreciation of our efforts.

I have only a few minor suggestions remaining, but otherwise am pleased with the changes made:

  1. Aim D could be revised slightly to be more clear and comprehensively explained, as it is still vague and did not seem to clearly lead to the logistic regression.

We have modified aim d following your suggestion:

“to ascertain the existence of a correlation between self-consciousness and self-conscious emotions and psychometric and clinical variables and identify the emotional variables that independently correlate with the diagnosis of AN”

  1. I noticed there were still some minor typographical errors throughout the manuscript, and some areas requiring further English language proofing to improve the clarity of expression. 

The draft has been deeply revised. You can follow throughout the script all the new changes highlighted in green. As we are waiting rev1 answer, we cannot upload a new draft. You can find the temporary draft with the corrections down here in the attachment box.